

# High frequency of transition to transversion ratio in the stem region of RNA secondary structure of untranslated region of SARS-CoV-2

Madhusmita Dash[1], Preetisudha Meher[1], Aditya Kumar[2], Siddhartha Sankar Satapathy[3] and Nima D. Namsa[2]

[1] Department of Electronics and Communication Engineering, National Institute of Technology Arunachal Pradesh, Jote, Arunachal Pradesh, India
[2] Department of Molecular Biology and Biotechnology, Tezpur University, Tezpur, Assam, India
[3] Department of Computer Science and Engineering, Tezpur University, Tezpur, Assam, India

Corresponding authors
Madhusmita Dash,
madhusmita.dash81@gmail.com
Nima D. Namsa,
namsa@tezu.ernet.in,
ndnamsa12@gmail.com

## ABSTRACT

**Introduction**. The propensity of nucleotide bases to form pairs, causes folding and the formation of secondary structure in the RNA. Therefore, purine (R): pyrimidine (Y) base-pairing is vital to maintain uniform lateral dimension in RNA secondary structure. Transversions or base substitutions between R and Y bases, are more detrimental to the stability of RNA secondary structure, than transitions derived from substitutions between A and G or C and T. The study of transversion and transition base substitutions is important to understand evolutionary mechanisms of RNA secondary structure in the 5′ and 3′ untranslated (UTR) regions of SARS-CoV-2. In this work, we carried out comparative analysis of transition and transversion base substitutions in the stem and loop regions of RNA secondary structure of SARS-CoV-2.

**Methods**. We have considered the experimentally determined and well documented stem and loop regions of 5′ and 3′ UTR regions of SARS-CoV-2 for base substitution analysis. The secondary structure comprising of stem and loop regions were visualized using the RNAfold web server. The GISAID repository was used to extract base sequence alignment of the UTR regions. Python scripts were developed for comparative analysis of transversion and transition frequencies in the stem and the loop regions.

**Results**. The results of base substitution analysis revealed a higher transition (*ti*) to transversion (*tv*) ratio (*ti/tv*) in the stem region of UTR of RNA secondary structure of SARS-CoV-2 reported during the early stage of the pandemic. The higher *ti/tv* ratio in the stem region suggested the influence of secondary structure in selecting the pattern of base substitutions. This differential pattern of *ti/tv* values between stem and loop regions was not observed among the Delta and Omicron variants that dominated the later stage of the pandemic. It is noteworthy that the *ti/tv* values in the stem and loop regions were similar among the later dominant Delta and Omicron variant strains which is to be investigated to understand the rapid evolution and global adaptation of SARS-CoV-2.

**Conclusion**. Our findings implicate the lower frequency of transversions than the transitions in the stem regions of UTRs of SARS-CoV-2. The RNA secondary structures are associated with replication, translation, and packaging, further investigations are needed to understand these base substitutions across different variants of SARS-CoV-2.

## INTRODUCTION

Biological information is stored in RNA as a sequence of four bases A, U, G, and C, of which A and G are purines (R) (two-ring structure), and C and U are pyrimidines (Y) (one-ring structure). The nucleotide bases in the RNA sequence tend to form base pairs with the help of hydrogen bonds that lead to the folding of RNA, called the secondary structure, which consists of loop and stem regions of unpaired and paired bases, respectively. The three canonical base pairs in the RNA stem region are complementary A:U and G:C and non-complementary base pairs G:U. This R:Y base-pairing is vital to maintain a uniform lateral dimension along the stem structure. Secondary structures are essential for RNA function; typical examples are the tRNA gene cloverleaf structure and stem-loop motif of rho-independent transcription termination site in many prokaryotes (*Kriner & Groisman, 2017*). The conserved coronavirus stem loop structures have been reported to perform functional roles in viral replication and RNA synthesis pathways (*Stammler et al., 2011*; *Yang & Leibowitz, 2015*).

Though in a shallow frequency, one base can replace any of the three other bases in a sequence. These changes in a sequence due to base replacement are called as substitution mutations. Base substitutions between A and G or C and T/U are transitions, while base replacements between R and Y bases are transversions. Out of the twelve possible base substitutions, eight are transversions (*tv*) (R → Y; Y → R), and four are transitions (*ti*) (R → R, Y → Y) (Fig. 1). If base substitutions occur randomly, the expected *ti/tv* ratio should be around 0.5 in any genome sequence. However, the observed ratio is usually 2.0 or more in any genome. The four-time higher observed *ti/tv* ratio, than the expected ratio, suggests that transitions are more acceptable than transversions in DNA sequences (*Lyons & Lauring, 2017*; *Stoltzfus & Norris, 2016*). The bias in *ti* over *tv* in genomes has been known since the early 1980s from the comparative studies of homologous DNA sequences of phylogenetically close species (*Gojobori, Li & Graur, 1982*; *Wu & Maeda, 1987*).

The higher frequency of the *ti* substitutions *versus* the *tv* substitutions can be explained from both selection and mutation point of views. Regarding selection mechanisms favoring *ti* over *tv*, the impact of amino acid replacement in protein structures has been suggested as the primary selection factor for higher *ti* frequency in protein-coding sequences. Single base substitution *tv* in triplet codons in the genetic code table produced more non-synonymous codons than the single base substitution *ti* (*Abdullah et al., 2016*). In a codon, purifying selection is more potent in non-synonymous sites than synonymous sites. This strategy of codon usage results in a decrease in *tv* compared to *ti* (*Eyre-Walker & Keightley, 1999*; *McDonald & Kreitman, 1991*; *Yang, 2007*).

Further, among the non-synonymous changes, *tv* results change one amino acid to a more dissimilar amino acid than *ti* results (*Vogel & Kopun, 1977*). According to mutation theory, *ti* is preferred over *tv* during DNA synthesis. This preference for *ti* is because the R: Y

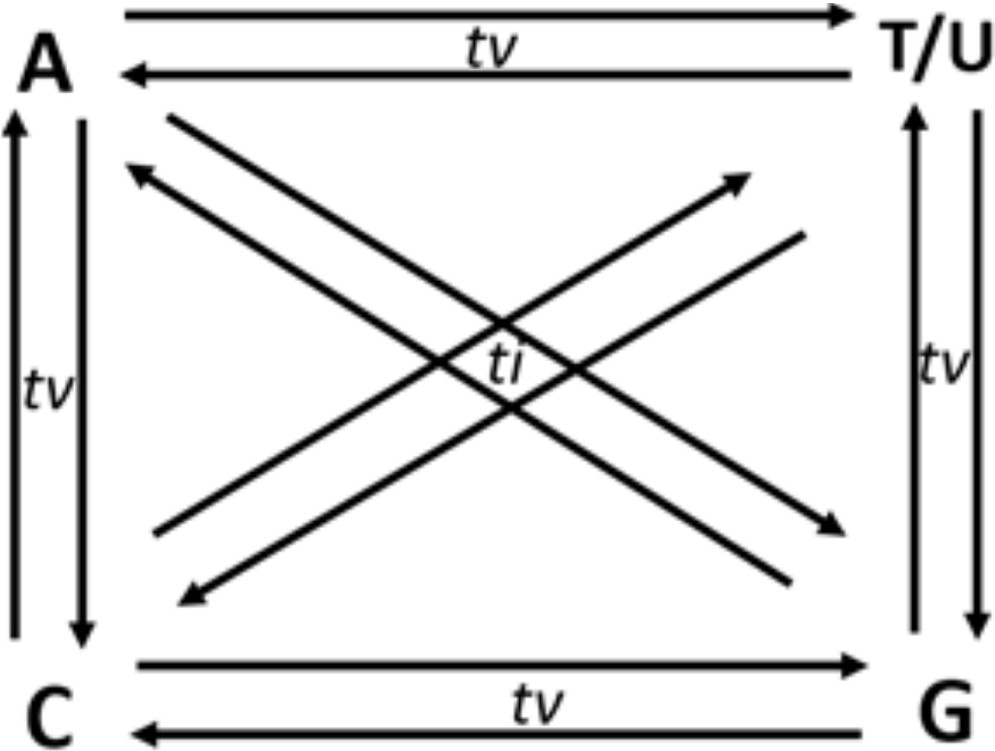

**Figure 1 Twelve possible directional mutations among four nucleotides.** Purine (R:A/G) → purine (R:A/G) or pyrimidine(Y:C/T) → pyrimidin(Y:C/T) base substitution mutations are called transitions (*ti*), whereas R →Y and Y →R substitutions are called transversions (*tv*). Out of the twelve possible directional substitutions, there are four types of transitions and eight types of transversions.

pairing is maintained to retain a regular DNA geometry; whereas this geometry is distorted by the R: R or Y: Y mispairing that is characteristic of *tv*. Therefore, *ti* and not *tv* is favored by DNA polymerase during DNA synthesis, since the R: Y mispairing responsible for *ti* is devoid of steric hindrances observed in the case of *tv*. In their model-building article, considering tautomerism and syn and anti-conformations of bases, *Topal & Fresco (1976)* argued that R:R mispairing is the primary mode of *tv* as opposed to Y:Y mispairing in DNA. This hypothesis was later proved to be correct by other researchers (*Fersht & Knill-Jones, 1981*; *Sinha & Haimes, 1981*). In addition, some of the frequently occurring DNA damage processes, such as cytosine deamination and ribonucleotide incorporation, favor *ti* in DNA (*Lewis et al., 2016*; *Schroeder et al., 2017*). Enzymatic processes of RNA-editing are among the other factors that can affect the rates of nucleotide mutations in RNA stems and loops motifs. Enzymes from ADAR family are known to bind stems and introduce A to I (finally, G) transitions in them. Enzymes from APOBEC superfamily are known to bind loops and introduce C to U (eventually, T in DNA) transitions (*Blanc & Davidson, 2010*; *Di Giorgio et al., 2020*; *Simmonds, 2020*). However, these processes do occur spontaneously as well. Oxidation of G and its incorrect repair is known as the mechanisms of G to T transversions (*Van Loon, Markkanen & Hübscher, 2010*). While, the C to U substitutions are unlikely to

affect the stability of stem-loop structures since U can pair at comparable efficiency with A or G nucleotides in RNA. Since this process is naturally more frequent in loops, stems may be protected from such transversions.

Transversions are likely to destabilize RNA secondary structures to a greater extent than transitions; therefore, transversion are reported to occur in lower frequency in the stem region (*Rossetti et al., 2015*). In Fig. 2, different versions of base substitutions in the stem and loop motifs of secondary structure are presented by using a hypothetical nucleotide sequence as an example. It is evident from Fig. 2 that both transition and transversion substitutions in the loop region have little impact on the secondary structure, as the mutated bases remain unpaired. However, substitutions in the stem region elicit a significant impact on the structure (Fig. 2). For example, C → U (*ti*) mutation results in G: U pairing in the stem region, whereas U → G (*tv*) and G → U (*tv*) mutations result in G:A and U:C mis-pairings, respectively in the stem region. The genome of SARS-CoV-2 contains a positive sense single-stranded RNA with 5′ capped and 3′ polyadenylated. Generally, in a genome with 5′ capping, translation initiation is believed to occur through a cap-dependent process. The role of the secondary structure of the UTR in determining efficiency of translation initiation in both cap-dependent and cap-independent mechanisms has been reported previously (*Babendure et al., 2006*). Recently, the base substitution pattern in the secondary structure was analyzed, applying experimentally determined stem-loop structures of 5′-UTR and 3′-UTR of SARS-CoV-2 (*Huston et al., 2021*; *Miao et al., 2021*) and a sequence alignment of the genome available in the GISAID database (*Shu & McCauley, 2017*). It is noteworthy at this juncture that transversions are known to induce secondary structure destabilization that might affect the efficiency of translation initiation. In the present investigation, higher frequencies of transitions were observed, compared to transversions, in the stem motifs than in the loops of RNA secondary structure of SARS-CoV-2.

## MATERIALS & METHODS

### Stem and loop annotations of SARS-CoV-2 reference genome

The reference SARS-CoV-2 isolate Wuhan-Hu-1 genome consisting of 29,903 bases (NC_045512.2) has been used for annotations of functional regions of the SARS-CoV-2 (*Wu et al., 2020*). For the base substitution study, the untranslated regions, 5′-UTR (base position 1 to 265) and 3′-UTR (base position 29,675 to 29,903) were considered. Unlike the protein coding gene sequences, these UTRs are devoid of any translational selection on codon usage bias (*Sharp & Li, 1986*). The stable secondary structure reported by *Miao et al. (2021)*, covered the 5′-UTR along with a portion of the nonstructural protein gene *nsp1*. This 5′-UTR secondary structure had been determined experimentally using radio-labeled transcript and RNase V1 enzymatic probing (*Miao et al., 2021*). The secondary structure reported by *Huston et al. (2021)* included a portion of the structural protein gene *N*, accessory protein gene ORF10, and the 3′-UTR. The well-defined stem-loop motifs (Table 1) of UTRs, except those paired with the bases of the neighboring coding regions were considered for the base substitution study. In subsequent sections in this article, these stem-loop structures are abbreviated as SL-I through SL-VII (Table 1).
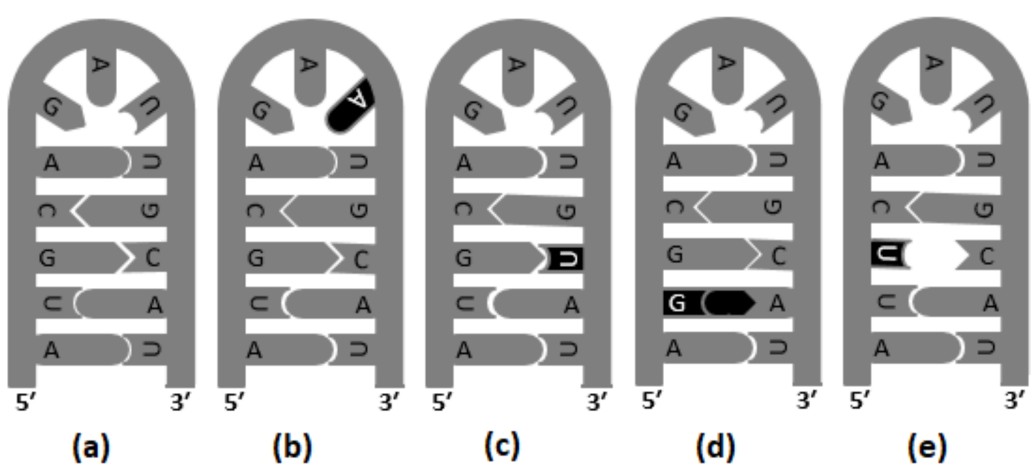

**Figure 2** **Base substitutions in the stem and loop motifs of the secondary structure.** The secondary structure of a hypothetical nucleotide sequence (5′-AUGCAGAUUGCAU-3′) and four possible scenarios of base substitutions in the secondary structure. The secondary structure of the hypothetical sequence (wild type) is given in (A). The four base substitution scenarios in the wild-type sequence are: U →A (*tv*) mutation in the loop region, which do not have any impact on the RNA secondary structure (B), C →U (*ti*) mutation resulting G:U pairing in the stem region (C), U →G (*tv*) mutation resulting G:A mis-pairing in the stem region (D), and G →U (*tv*) mutation resulting U:C mis-pairing in the stem region (E).

**Table 1** **Stem-loop structures in the SARS-CoV-2 (NC_045512.2) 5′-UTR and 3′-UTR.** The table presents seven well-defined stem-loop motifs considered for the base substitution study referred to by their short names SL-I through SL-VII. Out of these seven stem-loop motifs, the first six motifs SL-I through SL-VI are from 5′-UTR, and the last one SL-VII is from 3′-UTR.

| Location | Short name | Nucleotide sequence and secondary structures in dot-bracket notation |
|---|---|---|
| 7..33 | SL-I | GGTTTATACCTTCCCAGGTAACAAACC<br>(((((.(((((....)))))..))))) |
| 45..59 | SL-II | GATCTCTTGTAGATC<br>(((((.]]].))))) |
| 61..75 | SL-III | GTTCTCTAAACGAAC<br>((((..[[[[.)))) |
| 84..127 | SL-IV | CTGTGTGGCTGTCACTCGGCTGCATGCTTAGTGCACTCACGCAG<br>(((((((((.((.(((((.(((.....))).))))))).))))))))) |
| 130..146 | SL-V | TAATTAATAACTAATTA<br>((((((.....)))))) |
| 180..265 | SL-VI | CTTCTGCAGGCTGCTTACGGTTTCGTCCGTGTTGCAGCCGATCATCAGCACATCTAGGTTTCGTCCGGGTGT-<br><br>GACCGAAAGGTAAG<br>(((((..(((((((.(((((......))))))..))))))......)))((((((((.((......)))))))))))((((....)))))) |
| 29675..29830 | SL-VII | CAATCTTTAATCAGTGTGTAACATTAGGGAGGACTTGAAAGAGCCACCACATTTTCACCGAGGCCACGCG-<br>GAGTACGATCGAGTGT<br>ACAGTGAACAATGCTAGGGAGAGCTGCCTATATGGAAGAGCCCTAATGTGTAAAATTAATTTTAGTAGTG<br>((((((((((((((((((..((.(((((((((....((.((...((......))..))))<br>...))))).)))...))....)))...............)))))))))).........))).......)) |

Out of these seven stem-loop motifs given in Table 1, the first six motifs SL-I through SL-VI were from 5′-UTR, and the last motif SL-VII was from 3′-UTR. The stem-loop motifs in terms of dot-bracket notations given in Table 1 were visualized using RNAfold Forna software (http://rna.tbi.univie.ac.at/forna/) (*Gruber et al., 2008*; *Kerpedjiev, Hammer & Hofacker, 2015*; *Lorenz et al., 2011*; *Mathews et al., 2004*). The two stem-loop structures SL-II and SL-III were pseudoknotted. In SL-II, base positions 51, 52, and 53 in the loop regions were paired with positions 37, 38, and 39. Base positions 67, 68, 69, and 70 of the SL-III were paired with base positions 76, 77, 78, and 79. The secondary structures of the 5′-UTR and 3′-UTR considered in this study are given in Figs. S1 and S2, respectively. For the base substitution analysis, bases were categorized into two groups (i) paired or belonging to stem region and (ii) unpaired or belonging to loop regions. Out of the 204 positions analyzed in the 5′-UTR, 149 were categorized under the stem region, and the remaining 55 were categorized under the loop region. Out of the 156 positions analyzed in the 3′-UTR, 68 were categorized under the stem region, and remaining 88 were categorized under the loop region. In total, the percentage of bases considered under stem and loop were found to be 60.0 and 40.0, respectively.

## Retrieval and sequence alignment of UTRs of SARS-CoV-2 genome

In the present investigation, 46,076 high-coverage SARS-CoV-2 genome sequences were extracted on 24th July 2020 from the GISAID database (https://www.gisaid.org/) (*Shu & McCauley, 2017*). These genome sequences were sampled from patients, drawn from 95 countries, across the globe. These genome sequences represent the early stage of adaptation phase of SARS-CoV-2 pandemic in the human population. The downloaded genome sequences were processed to filter out sequences displaying size mismatch with the reference sequence NC_045512.2, including those with ambiguous nucleotides other than A/T/G/C. The final filtered set of 42,725 strains was retained and used to create a local BLAST database. Using 5′-UTR and 3′-UTR sequences of the reference genome (NC_045512.2) as query sequences, alignments of the two (5′- and 3′) untranslated regions were extracted from the local BLAST database, for base substitution analysis. In total, 4,049 sequences of the 5′-UTR and 2,811 strains of the 3′-UTR were available for the analysis. Alignments of the 5′-UTR and 3′-UTR sequences are given in Table S1. In addition to the above strains, dominant variants of Delta and Omicron strains reported in the GISAID database up to 24th September 2023, were also analyzed. After preprocessing, 8,227 sequences of the 3′-UTR and 7,020 sequences of the 5′-UTR, were available for base substitution analysis (Table S2).

## Identification of base substitutions in the stem and loop motifs of RNA secondary structure of SARS-CoV-2

To identify inter-species mutations in an alignment of homologous sequences of a few closely related species, researchers have often used methodologies based upon reconstructing a phylogenetic tree, and changes from ancestral sequences at various tree nodes (*Wu & Maeda, 1987*). Taking advantage of the large volume of SARS-CoV-2 genome sequences available in the public domain, a simple approach was employed in this

intra-species base substitution study. A consensus sequence considering the most frequent nucleotide at each position in the aligned sequences was generated. This consensus sequence was then compared with each sequence to identify base substitutions. Identification of base substitutions was carried out using the consensus sequence as shown in the hypothetical example (Fig. 3). The mutation frequencies were further normalized by dividing the total count of a given mutation, by the total number of nucleotides, in which the mutation occurred. For example, if C $\rightarrow$ U substitution count was found to be 1 in a sequence and the count of base C in that sequence was 10, then the normalized mutation frequency was calculated as 1/10 = 0.10. This consensus sequence-based method for estimating intra-species base substitutions is reported to be quite effective in mutation studies in bacteria genome tRNA gene secondary structures (*Sen et al., 2022*), estimating *dN/dS* for protein-coding genes (*Aziz et al., 2022*) and polymorphism analysis in intergenic regions (*Beura et al., 2023*). In subsequent sections, these normalized mutation frequency values are referred to as mutation frequencies. Each substitution in a sequence was further mapped to the stem-loop structure, and classified into loop and stem regions. If a pair of substitutions were observed in a paired position in the stem region, then they were considered compensatory; otherwise, they were designated non-compensatory substitutions. It is significant at this juncture, that the compensatory substitutions can be considered relatively older and more stable than the non-compensatory substitutions (*Higgs, 2000*). Therefore, the non-compensatory substitutions were found to be more frequent than the compensatory ones. Among all the stem-loop structures, only the non-compensatory substitutions were analyzed in this investigation Python scripts were written for identifying substitutions in the alignment of the SARS-CoV-2 secondary structures and their categorizations. The Python script, along with the executable and supporting stem-loop motif sequence files are available online for researchers in GitHub (https://github.com/MDash-NITAP/SLanalysis.git).

## RESULTS

In the beginning, a detailed study on base substitution in the SARS-CoV-2 genomes that represents the early stage of the pandemic period was carried out. Prior to identifying base substitutions in the stem-loop region, the frequency of twelve possible base substitutions in the SARS-CoV-2 genome was determined (Fig. S3). The base substitution frequencies of the four transitions A $\rightarrow$ G, G $\rightarrow$ A, C $\rightarrow$ U, and U $\rightarrow$ C was 0.170, 0.182, 0.505, and 0.162, respectively. The eight transversion frequencies were A $\rightarrow$ U (0.040), A $\rightarrow$ C (0.034), U $\rightarrow$ A (0.027), U $\rightarrow$ G (0.023), C $\rightarrow$ A (0.064), C $\rightarrow$G (0.014), G $\rightarrow$ U (0.230) and G $\rightarrow$ C (0.031). Transitions were generally more frequent than transversions in the genome, resulting in a *ti/tv* ratio equal to 2.20. Similar to the reported result from earlier studies (*Lewis et al., 2016*; *Matyášek & Kovařík, 2020*; *Simmonds, 2020*) C $\rightarrow$ U transition was found to be the most frequent base substitution followed by the transversion G $\rightarrow$ U. The total frequency of amino to keto base substitution M:(A/C) $\rightarrow$ K:(G/U) was 0.730, and the reverse amino to keto substitution K $\rightarrow$ M was 0.403, resulting in an overall frequency, that was skewed towards keto bases. Transition C $\rightarrow$ U is the most frequent substitution in the SARS-CoV-2 genome, and it constitutes 69.2% of the total M $\rightarrow$ K. This clearly

|  | 1 | 2 | 3 | 4 | 5 | 6 | 7 | 8 | 9 | 10 | 11 | 12 | 13 |
|---|---|---|---|---|---|---|---|---|---|---|---|---|---|
| Strain1 | A | U | G | C | A | G | A | U | U | G | C | A | U |
| Strain2 | A | G | G | C | A | G | A | A | U | G | C | A | U |
| Strain3 | A | U | G | C | A | G | A | U | U | G | U | A | U |
| Strain4 | A | U | G | C | A | G | A | U | U | G | C | A | U |
| Strain5 | A | U | G | C | A | G | A | A | U | G | C | A | U |
| Strain6 | A | U | G | C | A | G | A | U | U | G | U | A | U |
| Strain7 | A | U | G | C | A | G | A | U | U | G | C | A | U |
| Strain8 | A | U | G | C | A | G | A | U | U | G | U | A | U |
| Strain9 | A | U | U | C | A | G | A | U | U | G | C | A | U |
| Count$_A$ | 9 | 0 | 0 | 0 | 9 | 0 | 9 | 2 | 0 | 0 | 0 | 9 | 0 |
| Count$_U$ | 0 | 8 | 1 | 0 | 0 | 1 | 0 | 7 | 9 | 0 | 3 | 0 | 9 |
| Count$_G$ | 0 | 1 | 8 | 0 | 0 | 9 | 0 | 0 | 0 | 9 | 0 | 0 | 0 |
| Count$_C$ | 0 | 0 | 0 | 9 | 0 | 0 | 0 | 0 | 0 | 0 | 6 | 0 | 0 |
| Consensus Sequence 5'- | A | U | G | C | A | G | A | U | U | G | C | A | U -3' |
| Secondary Structure | ( | ( | ( | ( | ( | . | . | . | ) | ) | ) | ) | ) |

Mutations: U→G, G→U (at position 2–3), U→A (at position 8), C→U (at position 11)

**Figure 3** **Finding base substitutions in the secondary structure considering an alignment of nine hypothetical sequences containing thirteen nucleotides each.** The consensus sequence represents the most frequent nucleotide in each position in the alignment. The secondary structure of the consensus sequence is shown in dot-bracket notation. Any deviation at a particular position from the consensus sequence is considered as a base substitution, for example, there is a base substitution U → G at the 2nd position in the stem region and U → A at the 8th position in the loop region.

suggests that the method employed in the present study for estimating base substitutions is well correlated with the similar findings reported earlier (*Simmonds, 2020*; *Matyášek & Kovařík, 2020*).

## Higher mutation rate in the loop than the stem in the RNA secondary structure of SARS-CoV-2

Assuming the deleterious effect of mutations in the stem region, a comparative analysis between the rate of mutation in the stem and the loop region in the seven stem-loop motif sequences was carried out (Table 1). The unique mutations found in the stem-loop motif SL-I, SL-III, and SL-IV are shown in Fig. 4, and the remaining motif mutations are given in Table S3. In total, 360 base positions have been considered in the 5′-UTR and 3′-UTR, for the base substitution analysis. Of these base positions, 217 were categorized under

stem region, in which mutations were observed in 58 positions. On the other hand, 143 positions were categorized under the loop region, in which mutations were observed in 45 positions. Therefore, per position, the rate of mutation in the loop regions was found to be 0.315, whereas the rate was only 0.267 in the stem region. The observed difference in mutation counts between the stem and loop regions was found to be statistically significant ($p- < 0.01$). In addition, the most frequent base substitution C → U in the stem region was compared with the loop region (Fig. 4). The rate of base substitution C → U in the stem region (0.340), was found to be less than half of the rate in the loop region (0.690). This proportionately lower mutation rate in stem is in concordance with the notion, that the mutations in the stems destabilize RNA secondary structures and, therefore, they are counter-selected.

## Comparative analysis of transition and transversion in the stem and loop regions of RNA secondary structure of SARS-CoV-2

Twelve mutation frequencies considering all the stem-loop structures of the 5′-UTR and 3′-UTR were calculated and presented in Table 1. The mutation frequency values are shown in Fig. 5. Among the four transitions, C → U exhibited the highest frequency (0.364), which was more than the sum of the remaining three transitions U → C (0.132) G → A (0.126), and A → G (0.080). G → U (0.299) was the most frequent among the transversion mutations. This transversion value was even more than the transitions U → C, G → A, and A → G. This higher transversion frequency of G → U is consistent with earlier reported mutation patterns across different functional regions in SARS-CoV-2 genome. C → A (0.061) transversion was the next, followed by similar frequencies of A → U (0.050) and U → A (0.047), and G → C (0.046). Whereas, A → C (0.040), U → G (0.038), and C → G (0.024) were among the least frequent transversions. In total, transition and transversion frequencies were 0.704 and 0.605, respectively, resulting in a *ti/tv* ratio of 1.164, which was in accordance with the expected mutation pattern of the whole genome of the virus.

## The transition to transversion ratio in the stem is higher than that in the loop region of RNA secondary structure of SARS-CoV-2

Considering the differential impact of base substitutions in the stem and loop regions, base substitutions were calculated separately in the stem and loop regions (Fig. 5). Though the size of the stem region was larger than the loop region, the number of substitutions in the stem region was lower than the loop region. The base substitution rate in the stem region was 0.28, whereas the same was 0.36 in the loop region. This lower rate of mutations in the stem region suggested that the stem region is more conserved in comparison to the loop region. In general, transition mutations were more frequent than the transversions in stem regions as well as in loop regions (Fig. 5). Among the transitions in the stem region, C → U was the most frequent substitution with a frequency of 0.294, followed by U → C, G → A, and A → G with frequencies of 0.197, 0.121, and 0.056 respectively. G → U substitution was the most frequent transversion, with a frequency of 0.224, that was comparable with the C → U transition. Among the substitutions in the loop region, the C → U transition had the highest frequency (0.484), followed by the transversion G → U (0.448). The two

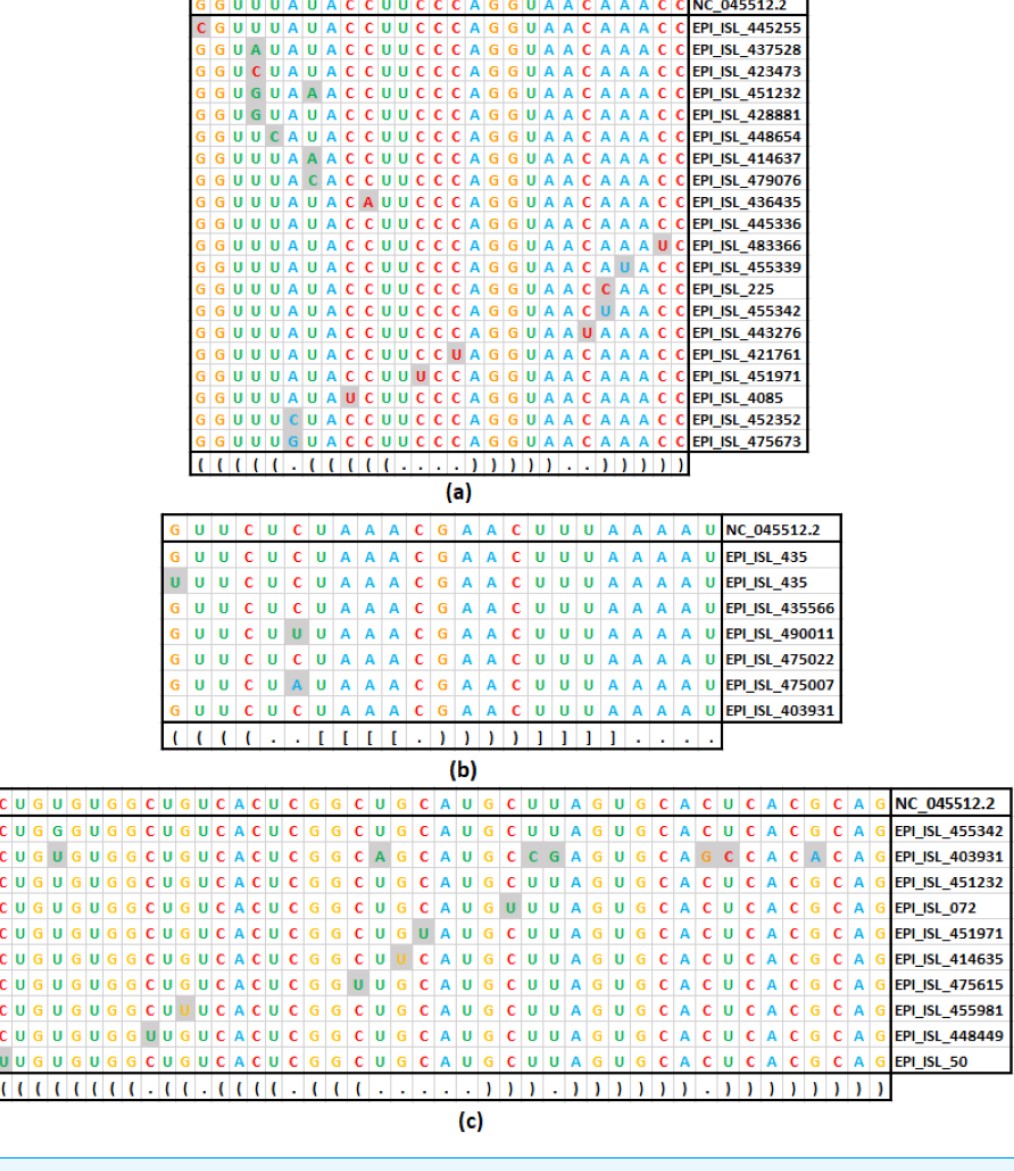

**Figure 4 Base substitution in three SARS-CoV-2 secondary structures.** (A): SL-I, (B): SL-III and (C): SL-IV of 5′-UTR. In each figure, the first row presents the reference sequence from NCBI of the structure, followed by an alignment of unique sequences with mutations considered in this study. The mutated base positions are shaded. The secondary structure stem-loop motif is given in the last row in dot-bracket notation.

other transitions G → A and A → G were third and fourth in order of frequencies with values of 0.138 and 0.109, respectively. Interestingly, the transition U → C frequency was very low (0.044) compared to the other three transitions in the loop region. The other transversion frequency values were within the range of 0.103 (G → C) to 0.022 (U → G).

Considering these mutation frequencies, transition to transversion ratio (*ti/tv*) in the stem and loop regions was calculated (Fig. 6). In the stem region, total transition and transversion frequency were found to be 0.667 and 0.453, respectively, resulting *ti/tv* values

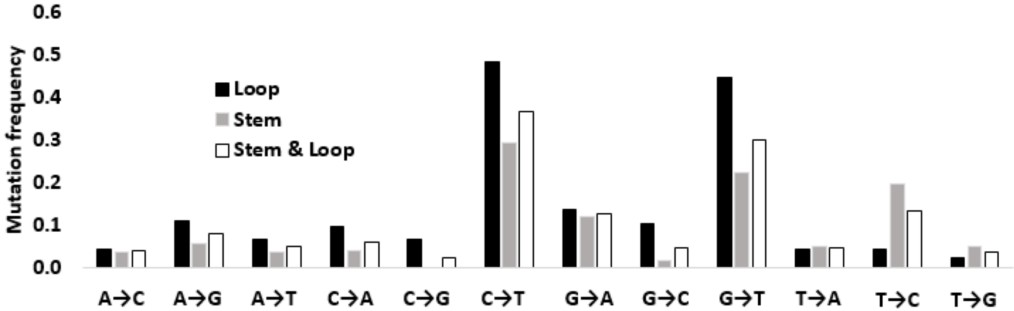

**Figure 5** **Mutation spectra in SARS-CoV-2 secondary structures.** Mutation spectra in the loop region stem region and combining both the regions of the secondary structures in the 5′-UTR and 3′-UTR of SARS-CoV-2 sequenced during the early phase of the pandemic period. The height of the vertical bars in the *Y*-axis represents twelve directional mutation frequency values in the stem and loop regions. The *X*-axis represents twelve mutations.

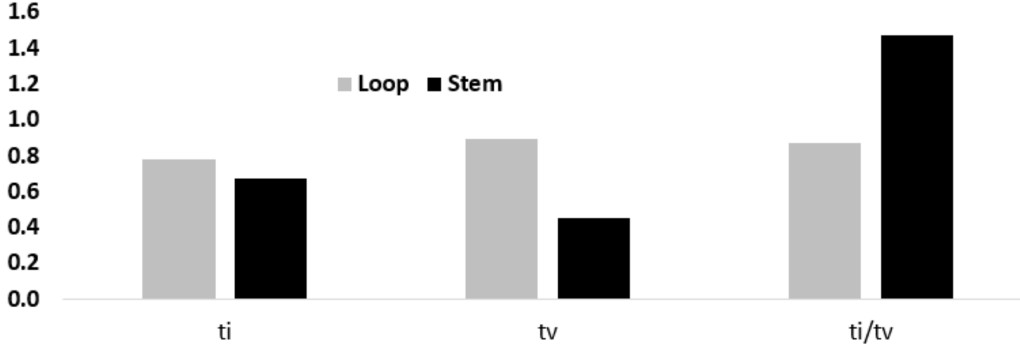

**Figure 6** **Higher *ti/tv* value in the stem region compared to the loop regions in SARS-CoV-2 UTR secondary structure.** Transition and transversion frequencies and the ratio *ti/tv* in the loop and stem regions of the secondary structures in the 5′-UTR and 3′-UTR of SARS-CoV-2. The height of the vertical bars in the *Y*-axis represents transition(*ti*) and transversion(*tv*) frequencies and the ratio *ti/tv* in the stem and loop regions.

of 1.472. In the loop region, total transition and transversion frequency were 0.775 and 0.888, respectively, resulting in *ti/tv* value of 0.872. The higher *ti/tv* value in the stem region, compared to the loop regions, was an outcome of the lower frequency of the transversion in the stem regions than the loop regions. This result suggested a deleterious effect of transversions in the stem region implicating that transversions might influence secondary structure of RNA.

In order to obtain statistical support for the higher ti/tv ratio in the stem regions, a Spearman rank correlation study between the twelve substitution frequencies in the stem and loop regions was done. When all the 12 mutation frequencies were considered, the Spearman rank correlation coefficient ($\rho$) was 0.510, suggesting that the order of the

frequency values in stem and loop were similar. Yet, the frequency of U → C transition in the stem and loop region ranked the third and ninth respectively, suggesting U → C transition was fairly accommodated in the stem region without distorting the secondary structure. In contrast, the frequency of G → C and C → G transversions in the stem region exhibited the lowest values, whereas in the loop region, they displayed the fifth and the eighth highest values, suggesting that the avoidance of G → C and C → G transversions was stronger in the stem region in comparison to the loop region. However, higher frequencies of G → U transversion were observed both in stem and loop regions.

### Analysis of base substitutions in the RNA secondary structure of Delta and Omicron variants of SARS-CoV-2

The base substitution analysis was extended to the stem and loop motifs of the UTRs of Delta and Omicron variants of SARS-CoV-2 (Fig. S4 ). In general, mutation frequency in the loop region was found to be with higher (2.537) in comparison to the stem region (2.131), indicating the differential role of secondary structure on base substitution. However, transition to transversion ratio ($ti/tv$) values in the stem and loop region were found to be similar. In the stem region, $ti$ value was 1.135 and $tv$ value was 0.996, resulting $ti/tv$ value of 1.140. In the loop region, the $ti$ value was 1.348 and, the $tv$ value was 1.189 resulting $ti/tv$ of 1.133. For understanding the similar $ti/tv$ ratios in the stem and the loop regions, a further Spearman rank correlation study between the twelve substitution frequencies was carried out. When all the 12 mutation frequencies were considered, Spearman rank correlation coefficient ($\rho$) was found to be 0.79 as expected. The C → T and G → A transitions and G → T transversion were with top three ranks in both stem and loop region. However, the notable differences in rank values of the base substitutions in stem and loop regions were as follows. The frequency of transitions A → G, and transversions G → C and C → G were with higher rank in the loop region compared to the stem region. In contrast, T → A and A → C transversions were with lower rank in the loop region compared to the stem region (Fig. S4).

## DISCUSSION

The large volume of genome sequence data generated since the outset of the SARS-CoV-2 pandemic provided a unique opportunity to investigate the long-term evolution of this virus. In this work, the patterns of base substitutions between the stem and loop motifs have been investigated using the experimentally determined secondary structures of 5′-and 3′-UTRs. These well-folded RNA structures of the SARS-CoV-2 genome are reported to be conserved across beta coronaviruses, which is important for the replication, translation, and packaging of the virus (*Jonassen, Jonassen & Grinde, 1998*; *Huston et al., 2021*; *Vora et al., 2022*). These structural features of 5′-UTR and 3′-UTR are also associated with viral infection (*Verma et al., 2021*) and therefore are an attractive target for designing anti-viral therapeutic agents (*Robertson et al., 2005*).

Intra-strand base pairing is important for the stability of the functionally significant secondary structure. Though the RNA transcripts of the SARS genome are known to have well-defined secondary structures, the role of base substitutions on the stability of RNA

secondary structure is yet to be explored adequately in the SARS-CoV-2 genome. The availability of secondary structure information motivated the present study to estimate and analyse transition and transversion substitutions in the UTRs of SARS-CoV-2. In the comparative study of *ti* and *tv* between loop and the stem motifs among strains sequenced during the early stage of the pandemic period, the stem region *ti* value was observed to be proportionately higher than *tv* when compared with the loop region. Transversion substitutions are known to destabilize the secondary structure of RNA; consequently, the lower frequencies of transversion mutations obtained in the stem regions, imply that transitions are accommodated to confer structural stability of UTR region in SARS-CoV-2. In contrast to early virus variants, the differential pattern of ti/tv values between stem and loop regions was not observed among the more advanced Delta and Omicron variants. It is possible that as the virus evolves, mutations become fixed and therefore the number of fixed mutations is higher in late variants than in early variants. The character of fixed variants may be different from the general mutation trend since it is purely driven by selection. SARS-CoV-2 is prone to accumulate rapid mutations in response to adaptation to a new human host leading to the emergence of newer variants over the period of time (*Pachetti et al., 2020*). The presence of higher number of fixed mutations across the genome of SARS-CoV-2 variants that dominated the later phase of the pandemic are key factor to the evolutionary dynamics of this rapidly mutating virus (*Kumar et al., 2022*; *Shah & Woo, 2022*; *Panja et al., 2023*). Further investigation is required to carry out detailed understanding of the observed mutations between the early phase variants and the late phase SARS-CoV-2 variants.

The 5′-UTR stable structures proximal to the AUG start codon and the UTR was reported to be highly conserved among SARS-CoV-2 genomes (*Miao et al., 2021*). The protein synthesis in SARS-CoV-2 was reported to begin *via* an unusual cap-dependent mechanism (*Conde et al., 2022*). The 5′-UTR contains signals for translation initiation. Interestingly, the 3′-UTR is known to regulate mRNA localization, and stability. In neurons, 3′ UTRs are well known to regulate local protein synthesis in dendrites and synapses (*An et al., 2008*; *Martin & Ephrussi, 2009*). In addition, 3′ UTRs can establish 3′ UTR-mediated protein–protein interactions, and thus can transmit genetic information encoded in 3′ UTRs to proteins (*Mayr, 2019*). It is noteworthy that the frequency of G → U transversion in the SARS-CoV-2 genome is very high, possibly because nucleotide base G gets oxidized to 8-oxoguanine or 8-nitroguanine in the oxidative environment (*Van Loon, Markkanen & Hübscher, 2010*; *Graudenzi et al., 2021*). The single-stranded RNA genome of SARS-CoV-2, may be highly prone to oxidative deamination of cytosine and guanine bases, as compared to double-stranded RNA and DNA viruses (*Sanjuan & Domingo-Calap, 2016*). Further investigation is needed to study the impact of high G → U transversion on the secondary structure of RNA. In this context, it will be interesting to investigate the role of transition and transversion substitutions in the stem-loop regions of RNA secondary structure of SARS-CoV-2 variants. In conclusion, our findings from this *in silico* study suggest that substitutions that negatively impact the secondary structure of RNA are not accommodated due to reduced fitness. Since transversions are more deleterious to secondary structures than transitions, their frequency in the virus genome is lower

than that of transitions. As the RNA secondary structures are associated with replication, translation, and packaging, it is important to understand these base substitutions across different variants of SARS-CoV-2.

## ACKNOWLEDGEMENTS

All the authors thankfully acknowledge Prof. S.K. Ray, Department of Molecular Biology and Biotechnology, Tezpur University for critical discussion on the manuscript. Siddhartha Sankar Satapathy and Nima D. Namsa thank the Bioinformatics and Computational Biology Centre, Tezpur University for the use of the computing facility.

### Funding

The authors received no funding for this work. The National Network Project for ACTREC-TMC, Navi Mumbai, Department of Biotechnology, Govt. of India (No. BT/PR40231/BTIS/137/63/2023) and the Bioinformatics and Computational Biology Centre for Microbial Biodiversity in Assam and Arunachal Pradesh, Department of Biotechnology, Govt. of India (No. BT/PR40253/BTIS/137/52/2022) paid the APC for this article. The funders had no role in study design, data collection and analysis, decision to publish, or preparation of the manuscript.

### Grant Disclosures

The following grant information was disclosed by the authors:
The National Network Project for ACTREC-TMC, Navi Mumbai, Department of Biotechnology, Govt. of India: No. BT/PR40231/BTIS/137/63/2023.
Bioinformatics and Computational Biology Centre for Microbial Biodiversity in Assam and Arunachal Pradesh, Department of Biotechnology, Govt. of India: No. BT/PR40253/BTIS/137/52/2022.

### Competing Interests

The authors declare there are no competing interests.

### Author Contributions

- Madhusmita Dash conceived and designed the experiments, performed the experiments, analyzed the data, prepared figures and/or tables, authored or reviewed drafts of the article, and approved the final draft.
- Preetisudha Meher analyzed the data, authored or reviewed drafts of the article, and approved the final draft.
- Aditya Kumar analyzed the data, prepared figures and/or tables, and approved the final draft.
- Siddhartha Sankar Satapathy analyzed the data, prepared figures and/or tables, authored or reviewed drafts of the article, and approved the final draft.
- Nima D. Namsa analyzed the data, authored or reviewed drafts of the article, and approved the final draft.

## Data Availability

The data is available at GitHub and Zenodo:

–https://github.com/MDash-NITAP/SLanalysis.git.

–Dash, M. (2024). SLanalysis. In PeerJ (sla24.2.1). Zenodo. https://doi.org/10.5281/zenodo.10593954.

## Supplemental Information

Supplemental information for this article can be found online at http://dx.doi.org/10.7717/peerj.16962#supplemental-information.

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
