# Peer review of "High frequency of transition to transversion ratio in the stem region of RNA secondary structure of untranslated region of SARS-CoV-2"

_PeerJ, doi:10.7717/peerj.16962_

## Round 0.1 · original submission · Major Revisions

Your manuscript on the transition and transversion frequencies in the stem and loop regions of SARS-CoV-2 non-transcribed regions contains novel information on coronavirus evolution and could be considered for publication in PeerJ after in-depth review.

I agree with the two reviewers that while your study has managed to captivate our attention there are some reservations regarding the experimental design and analysis, as well as the lack of crucial citations in certain sections.

In addition to addressing each and every one of the comments of both reviewers, the presentation of the results should be simplified and their interpretation improved. The number of figures could also be reduced to facilitate reading. Finally, I recommend using a professional English language edition.

With such substantial improvements, the manuscript could become a valuable contribution to the field. If the authors are willing to address the issues identified and make such significant revisions, the manuscript could be reconsidered for publication.

**Language Note:** PeerJ staff have identified that the English language needs to be improved. When you prepare your next revision, please either (i) have a colleague who is proficient in English and familiar with the subject matter review your manuscript, or (ii) contact a professional editing service to review your manuscript. PeerJ can provide language editing services - you can contact us at [email protected] for pricing (be sure to provide your manuscript number and title). – PeerJ Staff

Reviewer 1 ·

Basic reporting

English could be improved - please minor comments in the report

Experimental design

The research questions are defined, methods used are adequate.

Validity of the findings

This ms describes the mutation analysis of SARS-CoV-2 coronavirus genome. Such studies are needed in order to better understand the evolution of this highly pathogenic virus. The authors focus on structural motifs located in its 5´ and 3´untranslated regions. It is reported that transversion substitutions are underrepresented in stems compared to loops. The methods rely on standard bioinformatic approaches including computations of secondary structures and analysis of variants. The experiments are well-controlled, data sound and results could be of interest for the specialist in the field. There are several interpretation issues which should be addressed.

Additional comments

Specific comments

1. Mutations in stems are generally more deleterious than mutation in loops. This is because mutations in stems destabilize secondary structures of RNA. Therefore, mutations in stems are counter selected. It is therefore surprising that 21 positions were categorized under the loop and 37 positions under the stems in 5-UTR (line 231). I feel that this somewhat puzzling observation deserves an explanation.

2. The text of the ms is too descriptive and would benefit from a more biological focus. For example, the authors found variation in UTR regions among the SARS-CoV-2 genomes affecting secondary structure of RNA. It is known that the SARS-CoV-2 evolved into several strains during the pandemics, called alpha, beta, …omicron (please the WHO nomenclature). It would be most interesting to see if these mutations are correlated with particular virus strain. Perhaps, information about the virus lineage could be incorporated in Figures 3-4.

3. Line 259. The sentence “Base substitution per nucleotide in the stem region was 0.28 whereas the same is 0.36 in the loop region.“ The sentence is unclear, please revise. Perhaps, authors meant base substitution rate?

4. The authors calculated transition to transversion ratios (ti/tv) in the stem and loop regions finding that the ti/tv ratio was higher in loops than stems. Ok. However, I wonder how this fits with known mutation spectra of the SARS-CoV-2 coronavirus. Given that the Cs residues are far most common targets of mutations it might be relevant to count Cs in stems and loops. This can relatively easily be counted from the Table 1 data. If frequencies in both regions are the same I would agree that there is a selection pressure avoiding transversions in stems. In contrast, higher frequencies of Cs in loops compared to stems would offer a relatively trivially explanation of unequal ti/tv distribution in loops/stems.

5. Line 324. Conclusions. The sentence “This in silico study on transition and transversion mutations in SARS-CoV-2 UTR regions provides evidence that the secondary structure influences base substitutions“. I think that this interpretation is a wrong and the results should be viewed from the evolutionary //Darwinian/ perspective. As with other mutations base substitutions are subject to natural selection. Substitutions which negatively impact the secondary structure of RNA (transversions) are lost due to reduced fitness. Since transversions are more deleterious to secondary structures than transitions their frequency in the virus genome is lower than that of transitions.

6. There is something wrong with Figure order. For example, Figure 5 is actually Figure 4, Figure 6 is also Figure 4. Please revise.

7. The number of Figures could be reduced. For example, data from Figs.5-6 could be merged.

8. It came in my mind that it would interesting to analyze the distribution of substitutions in loops/stems of a negative strand since although short-lived the negative strand is important for virus replication.

Minor issues

English should be improved at many places.

Line 285. “whereas in the loop region, it was in the ninth position..“, better reads: “ whereas in the loop region, it ranked ninth“

Reviewer 2 ·

Basic reporting

The manuscript titled "Higher Transition to Transversion Ratio in the Secondary Structure Stem Motifs than the Loops in the SARS-CoV-2 Untranscribed Regions" presents an analysis of nearly 43,000 SARS-CoV-2 sequences, narrowed down to 66 unique sequences, to calculate transition and transversion frequencies in the 3'-UTR and 5'-UTR regions. The authors distinguish between stem and loop regions in these non-coding regions. However, the paper suffers from various shortcomings that undermine its scientific merit.

The manuscript demonstrates good use of scientific language and English, making it easy to comprehend.

I would like to acknowledge the manuscript's well-written introduction. The authors have done an excellent job of providing a comprehensive overview of the research topic, outlining the significance of investigating transition and transversion frequencies in the stem and loop regions of SARS-CoV-2 untranscribed regions. The introduction effectively sets the stage for the study and engages readers from the outset.

However, to further strengthen the introduction and enhance its credibility, I recommend the inclusion of a few additional citations to support the background information presented. For instance, in lines 56-60, where the authors discuss the importance of understanding the structural features of UTR regions, referencing relevant studies could add weight to their statements and demonstrate the depth of existing research in the field.

The authors conclude that the transition frequency in stem regions is similar to that in the loop regions, while the transversion frequency in stem regions is half of that in the loop regions, resulting in a higher transition to transversion ratio (ti/tv) in the stem region. However, the discussion and conclusions are notably brief and fail to delve into the implications of the findings adequately.

Experimental design

The experimental design has several significant shortcomings that raise concerns. The analysis is restricted to sequences from late 2019 to July 2020, a period when only the Alpha variant was detected. Considering the emergence of numerous variants after 2021, this timeframe is insufficient to draw meaningful conclusions about SARS-CoV-2 variability. Additionally, the identification of only 66 unique strains with mutations further supports the inadequacy of the sample size.

The analyses conducted offer little novelty compared to previous publications on the subject, except for the focus on loop and stem regions. However, the authors overlook a crucial aspect in their assessment. In the loop region, while individual bases may not interact, they can interact with other bases along the sequence, contributing to the tertiary structure of UTR regions. This oversight undermines the credibility of their conclusions, assuming that loop regions do not involve base interactions.

Furthermore, it is essential to consider that changes in bases within loop regions can lead to new interactions via complementarity, potentially extending the stem. Regrettably, the authors do not account for these considerations. Basic calculations such as substitutions/site/year, commonly performed in this type of analysis, are conspicuously absent, depriving readers of broader insights and contextualization of this study among others in the field of SARS-CoV-2 research.

Moreover, the authors mention the use of Python scripts but do not provide a repository to validate their analyses. This lack of transparency makes the results non-reproducible, hampering scientific progress and further investigation by other researchers.

Minor changes: The figures should use uracil (U) instead of thymine (T) to accurately represent RNA sequences, as SARS-CoV-2 is an RNA virus. If software limitations prevent using uracil, the authors should clearly explain their approach to mitigate potential confusion. Adhering to established standards will enhance the manuscript's scientific accuracy and credibility.

Validity of the findings

Due to the limitations in the experimental design, particularly the bias in using sequences likely belonging to the same SARS-CoV-2 variant that was no longer predominant after 2020, the results lack sufficient impact to offer valuable information to the scientific community. As a consequence, the findings fail to make a meaningful contribution to the current understanding of SARS-CoV-2 diversity and evolution.

In conclusion, the manuscript presents an interesting working hypothesis by comparing the stem and loop regions, shedding light on potential differences in transition and transversion frequencies within the SARS-CoV-2 untranscribed regions. The detailed introduction provides a strong foundation for the study, guiding readers through the context and rationale. However, the paper falls short in terms of its basic analysis and the failure to consider crucial factors that may affect the interpretation of the results. These limitations significantly undermine the validity and significance of the findings. To warrant publication, substantial revisions addressing the outlined issues are necessary.

---

## Round 0.2 · Minor Revisions

I believe you have made significant improvement to the manuscript. However, there are some important issues that need to be addressed before acceptance. I agree with the reviewers in their comments. I would be willing to receive a new version of the manuscript and a point-by-point response letter to the reviewers’ concerns. Thank you and Happy New Year

Reviewer 1 ·

Basic reporting

no comment

Experimental design

no comment

Validity of the findings

no comment

Additional comments

I thank the authors for submitting a revised version of the ms and their responses to my comments. While I am satisfied with most improvements they made there are several mostly minor issues that should be addressed. These are listed as below. Please note numbering of lines is according to the track changes version.

1. Line 30 “we did” better “we carried out”
2. Line 47. “..lower frequency of transversions“ better “..low frequency of transversion…“ or “lower frequency of transversions than the transitions..“
3. Line 238. This is probably the most serious issue. I calculated the rate of mutations in the loop region to be 43/143 = 0.301 and not 0.315. I just wonder how this influence the statistical between the groups since the differences are relatively small – 0.301 for the loop versus 0.267 for the stem.
4. Line 261. The sentence “The transition to transversion ratio in the stem is more…“. Better “The transition to transversion ratio in the stem is higher…“.
5. Line 289. “for a better understanding”. I am unsure if calculation of the Spearman coefficient helps “better understanding”. The coefficient is used for statistical evaluation of data. I would replace the term “better understanding” with “In order to obtain statistical support for the higher ti/tv ratio…”
6. Line 265 “..the stem region was more than the loop region“. Better “the stem region was larger than the loop region“. Further„“substitution“ – use a plural.
7. Line 266. Please replace “less“ with “lower“.
8. Line 342. The sentence “…5ʹ-UTR and 3ʹ-UTR is known to be important for efficient translation initiation“. Certainly, 5’UTR contains signals for translation initiation. However, I am not convinced that this holds true for the 3‘-UTR. The region is more commonly recognized as being important for mRNA localization and mRNA stability. Please check, and if possible, provide a relevant reference (e.g. Mayr (2008) What are 3‘-UTR doing? Cold Spring Harb Perspect Biol, DOI: 10.1101/cshperspect.a034728).
9. Contrast to early virus variant the differential pattern of ti/tv values between stem and loop regions was not observed among the more advanced Delta and Omicron variants. This observation is interesting while it is not discussed. For example, I wonder whether this has something to do with divergence of later variants from the reference genome which is derived from one of the earliest Sars-Cov-2 strains. One can for example imagine that as the virus evolves mutations are fixed. The number of fixed mutations is likely to be higher in late than early variants. The character of fixed variants may be different from the general mutation trend since it is purely driven by selection. This is exactly what is seen in Delta and Omicron mutation patterns.

Reviewer 2 ·

Basic reporting

I sincerely appreciate the significant improvements made in the revised manuscript compared to the initial submission. The expanded analysis now provides a much more comprehensive understanding of the viral diversity within SARS-CoV-2. The enhanced scope allows for a more thorough examination of key aspects related to viral mutations. It is evident that the authors have invested considerable effort in addressing the earlier limitations, resulting in a more robust and insightful contribution to the field.

I commend the authors for their diligence in broadening the range of the analysis, and I believe these advancements significantly strengthen the manuscript's scientific merit. I look forward to providing more detailed feedback on specific sections to further refine and elevate the overall quality of the work.

Experimental design

I would like to commend the meticulous effort dedicated to making the experimental design transparent and accessible to the broader scientific community. The decision to share the code on GitHub is particularly commendable, and the accompanying detailed manual is clear and user-friendly. This approach not only enhances the reproducibility of the study but also fosters collaboration and knowledge exchange within the scientific community.

Furthermore, I appreciate the notable improvements in this section compared to the previous manuscript. The incorporation of analyses related to new variants, despite the associated increased workload, demonstrates a commitment to staying at the forefront of research in the field. This expanded focus on newly emerging variants significantly contributes to the relevance and impact of the study.

Validity of the findings

I find the exploration of the Ti/Tv ratios across different variants to be particularly intriguing. The dynamic nature of these ratios, especially when correlated with specific variants, adds a layer of depth to the study's findings. It is a valuable and interesting observation that could have significant implications, particularly in the context of future research. Tracking these trends over time for each variant may provide insights into the stability or variability of these ratios, offering a unique perspective on the evolutionary dynamics of SARS-CoV-2.

Moreover, I would like to highlight the thoughtful emphasis on the UTR region. Differential focus on this region is a commendable aspect of the study, as mutations in untranslated regions can often carry understated yet critical functional implications. The recognition of the potential importance of mutations in these regions underscores the thoroughness of the analysis and contributes to a more comprehensive understanding of the genetic landscape of SARS-CoV-2.

Additional comments

I would like to make a minor note regarding the analysis of the delta and omicron variants and Suppl. 4 & 5. It has come to my attention that thymine (T) is being used as the nucleotide in this section. To maintain consistency with the rest of the manuscript, I suggest replacing T with U, considering the standard representation of RNA sequences.

Thank you for your attention to this matter, and I believe this adjustment will ensure coherence throughout the manuscript.

---

## Round 0.3 · Minor Revisions

I was about to accept the manuscript as I am happy with almost all the responses to the comments and edits to the text. I have not done so because I have a problem with the first of the two sentences on lines 352-354 that were included as a response to Reviewer 1's comment_item #9: "One can, for example, imagine that as the virus evolves the mutations are fixed. It is likely that the number of fixed mutations is greater in the late variants than in the early variants." The first one I think is too colloquial. Please replace the sentence with another one. My suggestion would be to merge the two into this one: "It is possible that as the virus evolves, mutations become fixed and therefore the number of fixed mutations is higher in late variants than in early variants."
As soon as I receive the corrected manuscript I will be ready to accept it.
Thank you very much for your excellent work.

---

## Round 0.4 · accepted · Accept

Thank you very much for your effort and work on the manuscript. Congratulations.